# Human genetic variants in SLC39A8 impact uptake and steady-state metal levels within the cell

Wen-An Wang[1], Andrea Garofoli[1], Evandro Ferrada[2,3,4], Christoph Klimek[1], Barbara Steurer[1], Alvaro Ingles-Prieto[1], Tanja Osthushenrich[5], Aidan MacNamara[5], Anders Malarstig[6], Tabea Wiedmer[1], Giulio Superti-Furga[1,7]

**The human *SLC39A8* (*hSLC39A8*) gene encodes a plasma membrane protein SLC39A8 (ZIP8) that mediates the specific uptake of the metals $Cd^{2+}$, $Mn^{2+}$, $Zn^{2+}$, $Fe^{2+}$, $Co^{2+}$, and $Se^{4+}$. Pathogenic variants within *hSLC39A8* are associated with congenital disorder of glycosylation type 2 (CDG type II) or Leigh-like syndrome. However, numerous mutations of uncertain significance are also linked to different conditions or benign traits. Our study characterized 21 *hSLC39A8* variants and measured their impact on protein localization and intracellular levels of $Cd^{2+}$, $Zn^{2+}$, and $Mn^{2+}$. We identified four variants that disrupt protein expression, five variants with high retention in the endoplasmic reticulum, and 12 variants with localization to the plasma membrane. From the 12 variants with plasma membrane localization, we identified three with complete loss of detectable ion uptake by the cell and five with differential uptake between metal ions. Further in silico analysis on protein stability identified variants that may affect the stability of homodimer interfaces. This study elucidates the variety of effects of *hSLC39A8* variants on ZIP8 and on diseases involving disrupted metal ion homeostasis.**

## Introduction

Metal ions are vital for ~80% of all cellular biochemical processes and play a role structurally and chemically in over a third of the organisms' proteome (1, 2). Many of these are essential metal ions that act as cofactors to a variety of proteins involved in metabolic pathways and transcriptional processes. Despite the importance of essential metals, their excessive concentration can result in cellular toxicity (3, 4, 5). Therefore, it is crucial to maintain metal ion homeostasis through catalysis, compartmentalization, and transport, as both deficiencies and overexposures have implications for health and disease.

The homeostasis of metal ions within cells is maintained by various proteins, such as the zinc-responsive (Zrt)– and iron-responsive transporter (Irt)–like protein (ZIP) family, a group of 14 cation transporters responsible for the transport of a variety of metal ions across cellular membranes (6, 7). Specifically, the human *SLC39A8* (*hSLC39A8*) gene encodes SLC39A8 (ZIP8), a plasma membrane (PM) transporter that is responsible for the influx of a variety of divalent cations into the cell. ZIP8 is broadly expressed across various tissues (https://www.proteinatlas.org/ENSG00000138821-SLC39A8/tissue), but is most highly expressed in the lung, duodenum, kidney, endometrium, placenta, breast, and tonsil. In addition, ZIP8 is present in specific cell types involved in the immune response within the central nervous system (8, 9, 10). The initial discovery of ZIP8 involved its role as a transporter of zinc ($Zn^{2+}$), but it is now known to play a role in the cellular uptake of a broader spectrum of divalent metals, including cadmium ($Cd^{2+}$), manganese ($Mn^{2+}$), iron (II) ($Fe^{2+}$), and cobalt ($Co^{2+}$) (6, 11, 12). Specifically, ZIP8 was proposed to be an electroneutral cotransporter of a divalent metal ion with two bicarbonate anions $(HCO_3^-)_2$ or one bicarbonate anion ($HCO_3^-$) and one selenite anion ($HSeO_3^-$) (13, 14). The uptake of $HSeO_3^-$ into the cell therefore also means that ZIP8 is an importer of selenium ($Se^{4+}$) (14).

Although the differential substrate affinity of ZIP8 and the metabolic consequences are complex and require further investigation, the functional transport of $Mn^{2+}$ is known to play a critical role in human pathophysiology. It has been shown that $Mn^{2+}$ is an essential cofactor for an array of enzymes, participating in many physiological processes, and is particularly important in the function and development of dopaminergic neurons (15, 16). Disruptions in $Mn^{2+}$ homeostasis have been associated with several diseases. Specifically, missense mutations in the *hSLC39A8* gene have been linked to two severe conditions associated with $Mn^{2+}$ deficiency: Leigh-like syndrome and congenital disorder of glycosylation (CDG). Leigh syndrome is an early-onset progressive neurodegenerative disorder characterized by defects in mitochondrial energy production caused by decreased activity of the key mitochondrial enzyme manganese superoxide dismutase (17,

[1]CeMM Research Center for Molecular Medicine of the Austrian Academy of Sciences, Vienna, Austria   [2]Centro Interdisciplinario de Neurociencia de Valparaíso, Facultad de Ciencias, Universidad de Valparaíso, Valparaíso, Chile   [3]Instituto de Neurociencia, Facultad de Ciencias, Universidad de Valparaíso, Valparaíso, Chile   [4]Instituto de Sistemas Complejos de Valparaíso (ISCV), Valparaíso, Chile   [5]Bayer AG, Pharmaceuticals, Wuppertal, Germany   [6]Pfizer Worldwide Research, Development and Medical, Stockholm, Sweden   [7]Center for Physiology and Pharmacology, Medical University of Vienna, Vienna, Austria

Correspondence: gsuperti@cemm.oeaw.ac.at

18, 19). CDG is a severe developmental disorder linked to a defect in the processing of protein-bound glycans. CDG type 2 (CDG type II) is the most common type associated with *hSLC39A8* genetic alterations and involves manganese-dependent $\beta$-1,4-galactosyl-transferase, an enzyme essential for protein glycosylation (20, 21). Moreover, reports on additional genetic variants implicate the transporter in several other diseases. The missense variant Ala391Thr (rs13107325) has been associated with extensive pleiotropic effects across several biological traits including blood pressure, body mass index, serum levels of N-terminal pro-B-type natriuretic peptide, total cholesterol, and triglyceride levels (10, 22). Furthermore, Ala391Thr is associated with several neuropsychiatric conditions such as schizophrenia and Parkinson's disease (10, 23), as well as cardiovascular diseases, liver disorders, and Crohn's disease (24, 25, 26). The extensive pleiotropic effects of this alteration may be attributed to the diverse cellular roles of $Mn^{2+}$, the wide range of substrates transported by ZIP8, and the combined role of the transporter within specific tissues. In addition to the transport of $Mn^{2+}$, the cellular uptake of $Zn^{2+}$ by ZIP8 is critical for immune cell function and it has been shown that $Zn^{2+}$ plays a pivotal role in the myeloid-mediated immune response to pneumococcal pneumonia and in lung epithelial cells during tumour necrosis factor–induced cytotoxicity (27, 28). Although the significance of *hSLC39A8* in both health and disease is widely recognized, the precise mechanism by which genetic variants cause the loss of protein function, and subsequently how this loss affects metal ion transport leading to disease, remains unclear.

The aim of this work was to study the effect of *hSLC39A8* genetic variants on the protein's impact on metal ion levels within the cell. We selected 21 genetic variants of the *hSLC39A8* gene, based on their relevance to human disease, and variant effect predictions (VEPs). We characterized the effect of these variants experimentally on protein expression, cellular localization, and cellular metal ion uptake, and computationally on dimerization stability. Ultimately, our goal is to highlight the impact of these mutations on the diverse spectrum of human diseases, which has the potential to hint at additional therapeutic avenues for *hSLC39A8* pathologies.

# Results

### Selection of variants for functional characterization

The selection of the variants to analyse in the study required a thorough survey and careful choice of different properties. Our study focused on the canonical isoform of the human *SLC39A8* (*hSLC39A8*) gene (GenBank Accession NM_001135146.2), which is composed of nine exons, is ubiquitously expressed, and encodes a transmembrane protein spanning 460 amino acids. We performed a literature review and compiled several *hSLC39A8* genetic variants from publicly available databases to assess their clinical and biological relevance (see the Materials and Methods section). As of January 2024, our search leads to 232 reported missense variants; of these, a large majority were unclassified variants, four were reported as pathogenic/likely pathogenic, four had conflicting interpretations of pathogenicity, five variants were classified as

benign/likely benign, and 24 were classified as variants of uncertain significance (VUS) (https://re-solute.eu/knowledgebase/gene/SLC39A8). Next, we performed in silico methods, by compiling publicly available VEP to assess the pathogenicity of all possible missense variants within the *hSLC39A8* coding region (Table 2).

For the final list of 21 variants that we chose, we incorporated several considerations (Tables 1 and 2). We included an early termination variant, Glu31* (V1), to serve as a loss-of-function genetic alteration control; 11 variants (V2-6, V8, V9, V13, V14, V16, and V19) that were reported from patients diagnosed with either CDG type II or Leigh-like disease; two variants (V7 and V21) reported with statistical association with non-pathogenic phenotypes; five VUS (V10-12, V15, and V17) with a potential pathogenic role based on the VEP scoring; and, lastly, two variants, Asn389Ser (V18) and Ala391Ser (V20), chosen because of their spatial proximity to the highly pleiotropic variant Ala391Thr (V19). To investigate the effects that the above-selected variants have on ZIP8 function, we generated stable HEK293 cell lines with the doxycycline-inducible overexpression of WT or variant ZIP8 (HEK SLC39A8^[WT] and HEK SLC39A8^[V1-V21]) (Fig 1A).

### Variant impact on cellular expression and localization

The ZIP8 transporter is predominately located at the PM, where it functions as an importer of divalent ions into the cell (8, 11). Mutations within the *hSLC39A8* gene that result in a non-functional transporter may also derive from its effect on cellular expression or PM localization. Therefore, as an initial assessment, we characterized the expression and subcellular localization of WT and variant ZIP8 through immunofluorescence imaging of HEK SLC39A8^[WT] and HEK SLC39A8^[V1-V21] cells (Fig 1A). WT and variant ZIP8 were visualized with the C-terminal HA tag and costained with markers for the nucleus, endoplasmic reticulum (ER), and PM (Figs 1B and S1). We further quantified the intensity of signal overlap between ZIP8 and the PM or between ZIP8 and the ER into Pearson's correlation coefficient ($r_{PM}$ and $r_{ER}$) for each variant (Fig 1C). As expected, ZIP8^[WT] colocalized adequately with both the PM ($r_{PM}$ = 0.56 ± SD = 0.10) and the ER ($r_{ER}$ = 0.68 ± SD = 0.09). We arbitrarily classified these variants into three categories based on the clustering of $r_{PM}$ versus $r_{ER}$ (Fig 1C): WT-like PM localization, high ER retention (relative to WT), and no expression. As ZIP8^[V1] is an early truncation mutation, it lacks the C-terminal HA tag for immuno-fluorescence detection and its localization was marked as inconclusive because of the lack of an appropriate conclusion. Based on the results, which we complemented with immunoblot analysis (Fig S2), we found three variants (V2, V5, and V19) that resulted in a lack of ZIP8 expression (Fig S1A), five variants (V3, V4, V6, V12, and V14) that lead to high ER retention (Fig S1B), and 12 variants (V7-11, V13, V15, V16-18, V20, and V21) that displayed PM localization comparable to WT (Fig S1C and D).

### Variant impact on the cellular uptake of $Cd^{2+}$, $Zn^{2+}$, and $Mn^{2+}$

The differential affinities of ZIP8 for the different metal substrates that it transports have been characterized in several studies (6, 34). Affinities for human ZIP8 have been reported for $Cd^{2+}$ (Km = ~1.1 ± 0.2 $\mu$M), $Mn^{2+}$ (Km = ~1.44 ± 0.39 $\mu$M), $Zn^{2+}$ (Km = ~1.3 ± 0.3 $\mu$M), and $Fe^{2+}$

**Table 1. Selection of 21 variants of the *hSLC39A8* gene for functional characterization.**

| ID | Variant | Reported phenotype | AF[a] | Clinical significance | Source | Literature |
|---|---|---|---|---|---|---|
| V1 | Glu31[a] | — | — | — | ENSEMBL ClinVar | — |
| V2 | Val33Met | CDG type II | $6.30 \times 10^{-4}$ <br> $6.41 \times 10^{-4}$ | Pathogenic | Literature | (6, 20, 21) |
| V3 | Gly38Arg | CDG type II | $3.22 \times 10^{-5}$ <br> $5.17 \times 10^{-5}$ | Pathogenic likely pathogenic | Literature ClinVar | (6, 20, 21) |
| V4 | Ser44Trp | Leigh syndrome-like | $-1.70 \times 10^{-5}$ | — | Literature | (29) |
| V5 | Cys113Arg | CDG type II | $-1.22 \times 10^{-6}$ | Uncertain significance | Literature ClinVar | — |
| V6 | Cys113Ser | Leigh syndrome | $3.98 \times 10^{-6}$ <br> $6.08 \times 10^{-7}$ | Likely pathogenic | Literature ClinVar | (18, 30) |
| V7 | Asp193Asn | Speed Amount of moving or speaking | $2.16 \times 10^{-5}$ <br> $1.82 \times 10^{-5}$ | — | — | — |
| V8 | Phe203Ser | SLC39A8 CDG | — | Uncertain significance | Literature ClinVar | — |
| V9 | Gly204Cys | CDG type II | $6.42 \times 10^{-5}$ <br> $5.90 \times 10^{-5}$ | Likely pathogenic | Literature | (6, 10, 12, 16, 21, 31, 32) |
| V10 | Trp305Cys | — | — | Uncertain significance | ClinVar | — |
| V11 | Thr308Met | — | $3.89 \times 10^{-5}$ <br> $2.55 \times 10^{-5}$ | Uncertain significance | ClinVar | — |
| V12 | Ile322Thr | — | $1.41 \times 10^{-5}$ <br> $1.16 \times 10^{-5}$ | Uncertain significance Likely pathogenic | ClinVar | — |
| V13 | Ser335Thr | CDG type II | — | — | Literature | (16, 21) |
| V14 | Ile340Asn | CDG type II | $-6.08 \times 10^{-7}$ | — | Literature | (16, 21) |
| V15 | Ile340Met | — | — | Uncertain significance | ClinVar | |
| V16 | Gly350Arg | Leigh syndrome−like | — | | Literature | (29) |
| V17 | Gln364Arg | — | $4.00 \times 10^{-6}$ <br> $6.08 \times 10^{-7}$ | Uncertain significance | ClinVar | — |
| V18 | Asn389Ser | — | $3.47 \times 10^{-4}$ <br> $3.50 \times 10^{-4}$ | Uncertain significance | ClinVar | — |
| V19 | Ala391Thr | Pleiotropic (>70 traits) | $4.51 \times 10^{-2}$ <br> $6.37 \times 10^{-2}$ | Benign | Open Targets Genebass Literature | (29, 33) |
| V20 | Ala391Ser | — | $3.99 \times 10^{-6}$ <br> $6.08 \times 10^{-7}$ | | | — |
| V21 | Leu449Phe | Volume of grey matter Alcohol use | $1.08 \times 10^{-2}$ <br> $1.25 \times 10^{-2}$ | Benign | Genebass GWAS | — |

[a]AF, allele frequency from gnomAD (first value) and RGC (second value).

(Km = ~23 ± 4.6 $\mu$M) (6, 34). To start the evaluation of the impact of ZIP8 mutations on the cellular uptake of metal ions, we first assessed $Cd^{2+}$ for its high affinity to ZIP8 (6, 12, 35). We used the FLIPR Calcium-5 Assay kit to measure $Cd^{2+}$ uptake (36, 37) and performed an initial $CdCl_2$ titration in HEK SLC39A8[WT] cells. We assessed an $EC_{50}$ of 2.64 $\mu$M (95% Cl: 2.13, 3.44) for the HEK SLC39A8[WT] cells (Fig 2A). Next, we measured $Cd^{2+}$ uptake in HEK SLC39A8[WT] and HEK SLC39A8[V1-V21] cells, using a set $CdCl_2$ concentration of 2 $\mu$M, based on the $EC_{50}$ (Fig 2B). Of the 21 variants, nine (V7, V9-11, V15, V17, V18, V20, and V21) exhibited WT-like cellular uptake of $Cd^{2+}$ and the remaining 12 variants (V1-6, V8, V12-14, V16, and V19) did not show any measurable response (Fig 2B, and Table 2). As expected,

although most variants with PM localization showed detectable $Cd^{2+}$ uptake, all variants resulting in no ZIP8 expression or high ER retention did not exhibit a response. Notably, only three variants with PM localization failed to show measurable $Cd^{2+}$ uptake (Table 2).

To determine whether variants with PM localization had an impact on substrate specificity, we measured the cellular uptake of $Zn^{2+}$ and $Mn^{2+}$ for the 12 WT-like PM localization variants (V7-11, V13, V15, V16, V17, V18, V20, and V21). As a negative control, we included the early termination variant Glu31* (V1). The cellular uptake of $Zn^{2+}$ was measured using a $Zn^{2+}$-sensitive cytoplasmic Zinpyr-1 dye (38, 39). To identify the best signal window of the $Zn^{2+}$ uptake between

**Table 2. Summary of results for the 21 variants of the *hSLC39A8* gene characterized within this study.**

| ID | Variant | AlphaMissense | Localization | Cd$^{2+}$ | Zn$^{2+}$ | Mn$^{2+}$ | Competition | Suggested reclassification |
|----|---------|---------------|--------------|-----------|-----------|-----------|-------------|----------------------------|
| V1 | Glu31* | (P) | — | No response | No response | No response | — | — |
| | | | | Decreased | No response | No response | | |
| V2 | Val33Met | 0.4699 (U) | NO | No response | — | — | — | Pathogenic |
| V3 | Gly38Arg | 0.8675 (P) | ER | No response | — | — | — | Pathogenic |
| V4 | Ser44Trp | 0.7008 (P) | ER | No response | — | — | — | Pathogenic* |
| V5 | Cys113Arg | 0.9414 (P) | NO | No response | — | — | — | Pathogenic* |
| V6 | Cys113Ser | 0.9571 (P) | ER | No response | — | — | — | Pathogenic |
| V7 | Asp193Asn | 0.2472 (B) | PM | As WT | Decreased | Decreased | As WT | Benign* |
| | | | | As WT | As WT | As WT | | |
| V8 | Phe203Ser | 0.8116 (P) | PM | No response | No response | No response | — | Pathogenic* |
| V9 | Gly204Cys | 0.1771 (B) | PM | As WT | As WT | As WT | As WT | Benign* |
| | | | | As WT | As WT | As WT | | |
| V10 | Trp305Cys | 0.8938 (P) | PM | As WT | As WT | Decreased | As WT | Likely pathogenic* |
| | | | | Decreased | Decreased | Decreased | | |
| V11 | Thr308Met | 0.3317 (B) | PM | As WT | As WT | Decreased | Zn$^{2+}$>Mn$^{2+}$ | Likely pathogenic* |
| | | | | As WT | As WT | Decreased | | |
| V12 | Ile322Thr | 0.8166 (P) | ER | No response | — | — | — | Pathogenic* |
| V13 | Ser335Thr | 0.5344 (U) | PM | No response | No response | No response | — | Pathogenic* |
| V14 | Ile340Asn | 0.9873 (P) | ER | No response | — | — | — | Pathogenic* |
| V15 | Ile340Met | 0.8372 (P) | PM | As WT | As WT | As WT | As WT | Benign* |
| V16 | Gly350Arg | 0.7856 (P) | PM | No response | No response | No response | As uninduced | Pathogenic* |
| V17 | Gln364Arg | 0.4003 (U) | PM | As WT | As WT | Decreased | As WT | Likely pathogenic* |
| | | | | As WT | Decreased | As WT | | |
| V18 | Asn389Ser | 0.0637 (B) | PM | As WT | Decreased | Decreased | As WT | Likely pathogenic* |
| | | | | Decreased | As WT | As WT | | |
| V19 | Ala391Thr | 0.2361 (B) | NO | No response | — | — | — | Pathogenic* |
| | | | | No response | No response | No response | | |
| V20 | Ala391Ser | 0.0833 (B) | PM | As WT | As WT | As WT | As WT | Benign* |
| V21 | Leu449Phe | 0.201 (B) | PM | As WT | As WT | As WT | As WT | Benign |

B, benign; P, pathogenic; U, ambiguous/uncertain; NO, no expression; ER, high endoplasmic retention; PM, plasma membrane localization; *—classification that is new from the previous clinical status; uptake assay results—first row; ICP-MS results (if performed)—second row.

ZIP8 variants, we measured the fluorescence of Zinpyr-1 in induced and uninduced HEK SLC39A8$^{WT}$ cells exposed to previously reported ZnCl$_2$ concentrations (40, 41) and determined 100 $\mu$M ZnCl$_2$ to be a suitable concentration (Fig 2C). As expected, the early truncation Glu31* (V1) variant did not result in Zn$^{2+}$ uptake (Fig 2D). Of the remaining 12 PM localizing variants tested, seven (V9-11, V15, V20, and V21) showed WT-like uptake, whereas three variants (V8, V13, and V16) did not have a measurable response to ZnCl$_2$, and two variants (V7 and V18) had diminished uptake activity (Fig 2D and Table 2).

For the measurement of the Mn$^{2+}$ uptake by the cell, we leveraged the cytotoxic properties of Mn$^{2+}$ overexpose to establish a cell viability assay as an indirect approach (3, 42). We first performed a MnCl$_2$ titration in induced and uninduced HEK SLC39A8$^{WT}$ cells and measured the percentage of cell survival after 48 h. We determined an LC$_{50}$ of 0.048 $\mu$M (95% Cl: 0.04 to 0.07) for HEK SLC39A8$^{WT}$ cells (Fig 2E). Next, we measured the survival of the variant-expressing cells exposed to several MnCl$_2$ concentrations and recorded the percentage of cell death at 0.1 $\mu$M of MnCl$_2$, a concentration selected for the best signal window for the Mn$^{2+}$ cell viability assay between ZIP8 variants (Fig 2F). In congruence with the Zn$^2$ uptake assay, the early truncation Glu31* (V1) variant did not show Mn$^{2+}$ uptake activity (Fig 2F and Table 2). Also consistent with the Zn$^{2+}$ uptake data, four variants (V9, V15, V20, and V21) were associated with WT-like uptake activity, whereas three variants (V8, V13, and V16) showed no response, and two variants (V7 and V18) showed decreased uptake (Fig 2F and Table 2). In contrast to the Zn$^{2+}$ uptake data, three variants (V10, V11, and V17) with WT uptake for Zn$^{2+}$ showed diminished uptake for Mn$^{2+}$ (Fig 2F and Table 2).

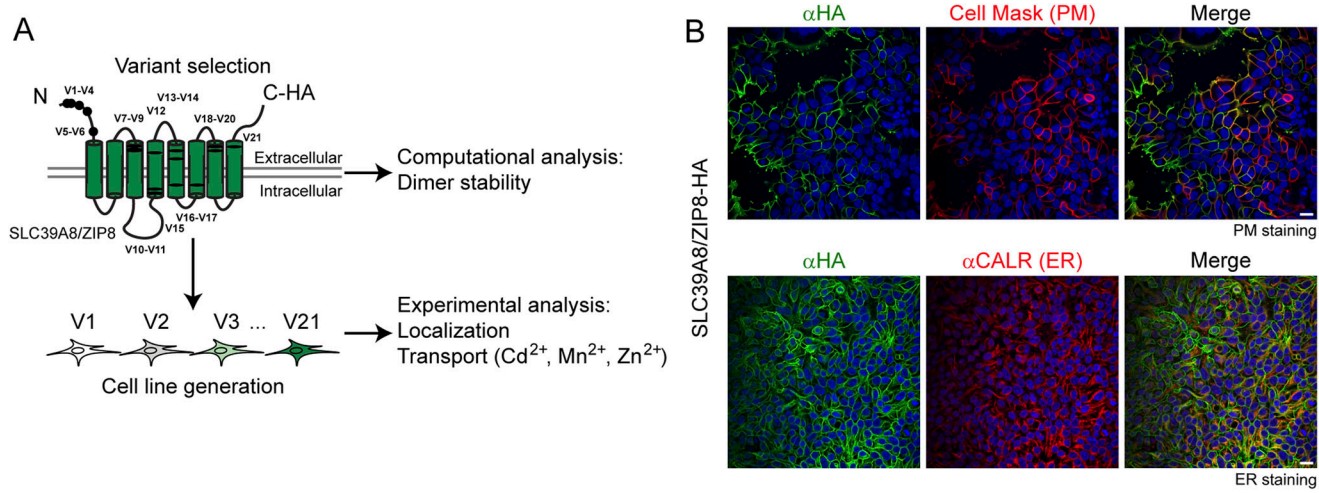

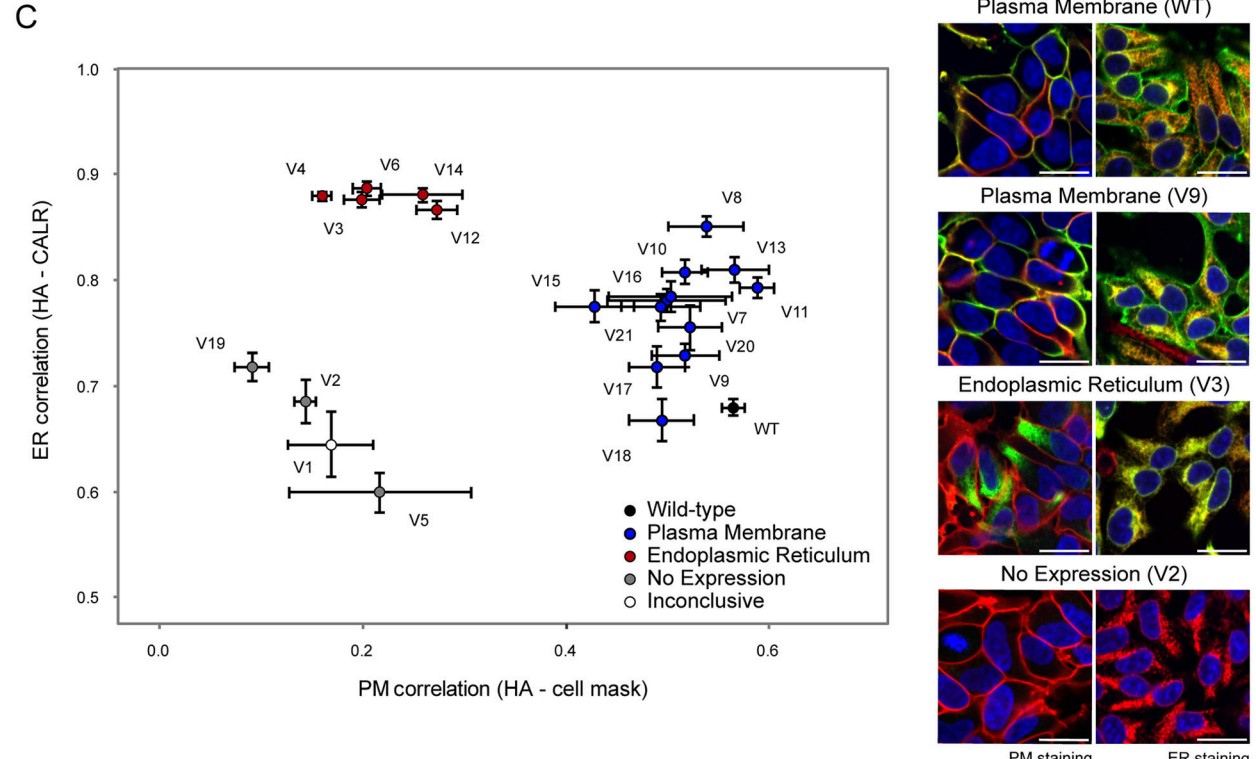

**Figure 1. Subcellular localization of ZIP8 variants.**
**(A)** Workflow schematic of the study on 21 single variants of SLC39A8, highlighted in black on the topology of the SLC39A8/ZIP8 transporter. **(B)** Representative immunofluorescence images of HEK SLC39A8[WT] cells stained with anti-HA ($\alpha$HA) antibody for ZIP8, cell mask for plasma membrane (PM), and anti-calreticulin ($\alpha$CALR) antibody for the endoplasmic reticulum (ER). Representative of 81–135 images. **(C)** Quantification of subcellular localization of ZIP8WT and ZIP8V1-21. A Pearson correlation coefficient was calculated between the immunofluorescence signals of each respective variant as detected by anti-HA antibody versus the PM (cell mask) or ER (anti-calreticulin) markers. Bars show the standard error of the mean over at least four replicates. Variants are grouped based on cluster formation with the colour indicating the impact of the variant on protein expression and localization. Representative images are presented in the right panel to display the different impacts on protein expression and localization (WT, ER, no expression).

## ICP-MS and Zn$^{2+}$ and Mn$^{2+}$ competition

To validate key findings from the uptake assays of Cd$^{2+}$, Zn$^{2+}$, and Mn$^{2+}$, we performed inductively coupled plasma mass spectrometry (ICP-MS) for a set of variants (V1, V7, V9, V10, V11, V17, V18, and V19) (Fig 3A–C). Consistent with the uptake results, we saw increased accumulation of intracellular Cd$^{2+}$, Zn$^{2+}$, and Mn$^{2+}$ for V7, V9, V10, V11, V17, and V18, and not for V1 and V19 (Fig 3A–C). Notably, there were

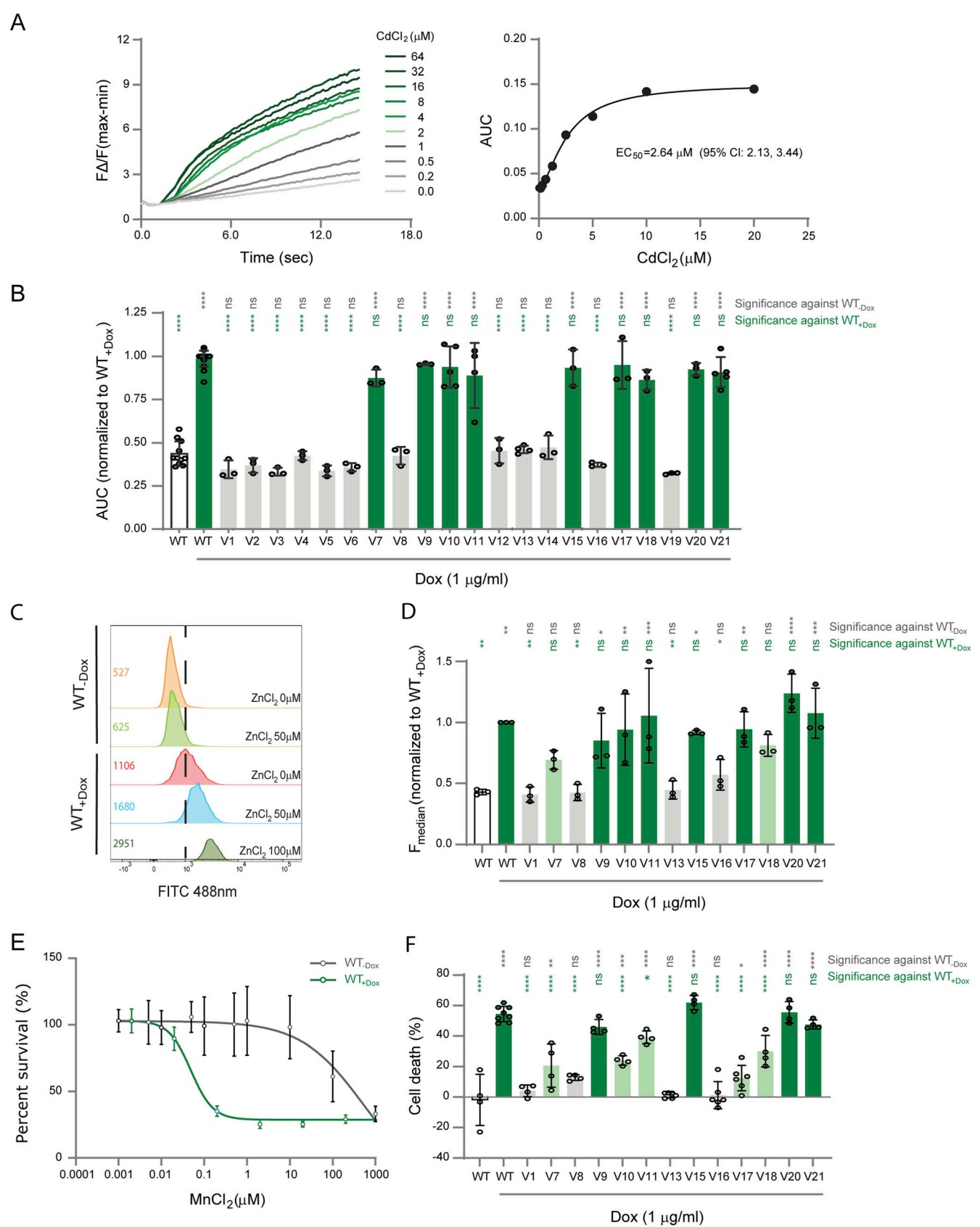

some discrepancies between the uptake assays and the ICP-MS results. Specifically, we did not see a decreased accumulation of $Zn^{2+}$ and $Mn^{2+}$ for V7 and V18, there was a decrease for all ions tested in V10, and there was a decrease in $Zn^{2+}$ but not in $Mn^{2+}$ for V17 (Fig 3A–C, Table 2).

Our uptake assays revealed three variants with WT-like cellular uptake for $Zn^{2+}$ but a lower cellular uptake for $Mn^{2+}$ (Fig 2D and F and Table 2). We therefore explored the impact of variants on substrate specificity between $Zn^{2+}$ and $Mn^{2+}$. We performed a competition assay using Zinpyr-1 to measure $Zn^{2+}$ uptake with a set of 40 $\mu M$ $ZnCl_2$, a concentration that provided the best signal window with the addition of $MnCl_2$, and a titration of $MnCl_2$ to determine the concentration of $Mn^{2+}$ required to interfere with $Zn^{2+}$ uptake. We selected all the variants with WT or decreased transport activity for both $Zn^{2+}$ and $Mn^{2+}$ and included a PM localizing, no response variant (V16) as a negative control. We quantified the $IC_{50}$ of $MnCl_2$ for each variant and compared it with the $IC_{50}$ values obtained in uninduced HEK SLC39A8$^{WT}$ cells (364.9 ± 216.9) and induced HEK SLC39A8$^{WT}$ cells (798.9 ± 159.4) (Fig 3D). As anticipated, V16 exhibited an $IC_{50}$ value of $MnCl_2$ statistically similar to that observed with no ZIP8 expression. $MnCl_2$ $IC_{50}$ values from eight variants (V7, V9, V10, V15, V17, V18, V20, and V21) showed no statistical difference compared with ZIP8$^{WT}$, suggesting that these variants did not alter substrate specificity (Fig 3D and Table 2). In contrast, V11 displayed a significantly higher $IC_{50}$ than ZIP8$^{WT}$, indicating increased uptake of $Zn^{2+}$ over $Mn^{2+}$ (Fig 3D and Table 2). To further elucidate the mechanistic basis for the observed differences, we investigated the effects of these variants on the structure of ZIP8, specifically in the WT dimer configuration.

## Structure prediction of ZIP8 suggests homodimerization

Several members of the ZIP family, including ZIP8, form multimers (43, 44, 45, 46). To investigate whether variants affect multimer formation, we used AlphaFold to predict and compare the structures of ZIP8 as a monomer versus a dimer, and as a dimer versus a trimer. Each modelling step was performed five times, and structural quality was assessed using in silico stability predictions and protein–protein interaction scoring methods. Building upon previous findings (8, 46, 47), our models revealed that the dimer exhibits significantly greater stability in comparison with the monomer configuration. In addition,

we used a protein–protein interaction score (pDockQ score) that accounts for the quality of the complex (48, 49). Such a score favoured the dimer over the trimer model. Moreover, the extracellular domain of the dimeric structure revealed high structural similarity to an experimentally characterized ZIP4 dimer (50). We therefore propose the most probable WT configuration of the functional ZIP8 protein of dimeric nature (Fig 4A).

The dimer formation suggests that some mutations that can be mapped within the interaction interface of the two ZIP8 monomers may affect stability of the dimer and lead to a non-functional transporter. To identify mutations of this type, we used in silico estimations of changes in thermodynamic stability upon a single amino acid substitution in the dimer versus the monomer configuration (Fig 4B). We reasoned that mutations causing a strong change in stability in the dimer, but not in the monomer, could reveal amino acids essential for the stability of the dimer. Using this strategy, we explored the upper fifth percentile of the most destabilizing amino acid substitutions, which amounted to 442 mutations with a ΔΔG larger than 2.4 kcal/mol, distributed over 105 positions. The analysis revealed clusters of positions more prone to dimer destabilization (Fig 4C). We found that at residues 29–36, 101–110, 193–209, 306–311, 344–351, and 397–409, as many as 5 to 10 different amino acid substitutions were destabilizing. Although none of our selected variants were the exact mutations in the upper fifth percentile list, a few variants were located within the destabilizing clusters. In particular, these included V1 (Glu31*) and V2 (Val33Met) in the first 29–36 cluster; V7 (Asp193Asn), V8 (Phe203Ser), and V9 (Gly204Cys) in the 193–209 cluster; V11 (Thr308Met) in the 306–311 cluster; and V16 (Gly350Arg) in the 344–351 cluster. Furthermore, V3 (Gly38Arg) was two residues away from Val36, with a peak number of eight destabilizing mutations, and V11 (Thr308Met) was three residues away from Asp311 with a peak number of 16 destabilizing mutations. Our analysis therefore identified the protein residues likely to be essential for ZIP8 dimerization and attributed potential variant effects within our study to dimer destabilization.

# Discussion

A major challenge in modern biomedicine arises from the necessity to decipher the functional significance of the numerous possible

**Figure 2. Transport activity of ZIP8 variants.**
**(A) (Left)** Representative recordings of FLIPR Calcium-5 dye fluorescence in response to $CdCl_2$ (0–64 $\mu M$) addition in HEK SLC39A8$^{WT}$ cells. The fluorescence signal was normalized to basal, before substrate addition (FΔ/F). **(Right)** The area under the curve (AUC) from the recordings (left) is averaged and plotted against the $CdCl_2$ concentrations to determine an $EC_{50}$ of 2.64 $\mu M$ (95% Cl: 2.13, 3.44). Average of two biological replicates. **(B)** Cadmium transport activity in HEK SLC39A8$^{WT}$ and HEK SLC39A8$^{V1-21}$ cells. The activity defined by the area under the curve (AUC) from the variant-expressing cells was normalized to that of HEK SLC39A8$^{WT}$ cells induced with doxycycline (WT$_{+DOX}$). Green indicates WT-like activity, and grey indicates no activity. At least three biological replicates were performed per variant. An ordinary one-way ANOVA test was used to determine statistical significance, and multiple comparisons were analysed against WT$_{-DOX}$ (grey) and WT$_{+DOX}$ (green) (ns, not significant; ****$P < 0.0001$).
**(C)** Representative fluorescence (FITC 488 nm) signal of Zinpyr-1 in response to $ZnCl_2$ (0–100 $\mu M$) in addition to WT$_{+DOX}$ and WT$_{-DOX}$ cells. Mean fluorescence intensities are indicated on the left of the plot, and the dotted line indicates the mean intensity of WT$_{+DOX}$ cells treated with 0 $\mu M$ of $ZnCl_2$. Representative of two biological replicates. **(D)** Zinc transport activity in HEK SLC39A8$^{WT}$ and HEK SLC39A8$^{V1-21}$ cells. The activity defined by median fluorescence (F$_{Median}$) from the variant-expressing cells was normalized to that of HEK SLC39A8$^{WT}$ cells induced with doxycycline (WT$_{+DOX}$). Green indicates WT-like activity, light green indicates decreased activity, and grey indicates no activity. Three biological replicates were performed per variant. An ordinary one-way ANOVA test was used to determine statistical significance, and multiple comparisons were analysed against WT$_{-DOX}$ (grey) and WT$_{+DOX}$ (green) (ns, not significant; *$P < 0.0332$; **$P < 0.0021$; ***$P < 0.0002$; ****$P < 0.0001$). **(E)** Per cent survival (%) of HEK SLC39A8$^{WT}$ cells, induced and non-induced with doxycycline, in response to $MnCl_2$ (0–1 mM) treatment for 48 h. Bars show the standard error of the average over two to four biological replicates.
**(F)** Manganese transport activity in HEK SLC39A8$^{WT}$ and HEK SLC39A8$^{V1-21}$ cells. The transport activity defined by the percentage of cell death is graphed. Green indicates WT-like activity, light green indicates decreased activity, and grey indicates no activity. Three biological replicates were performed per variant. An ordinary one-way ANOVA test was used to determine statistical significance, and multiple comparisons were analysed against WT$_{-DOX}$ (grey) and WT$_{+DOX}$ (green) (ns, not significant; *$P < 0.0332$; **$P < 0.0021$; ***$P < 0.0002$; ****$P < 0.0001$).

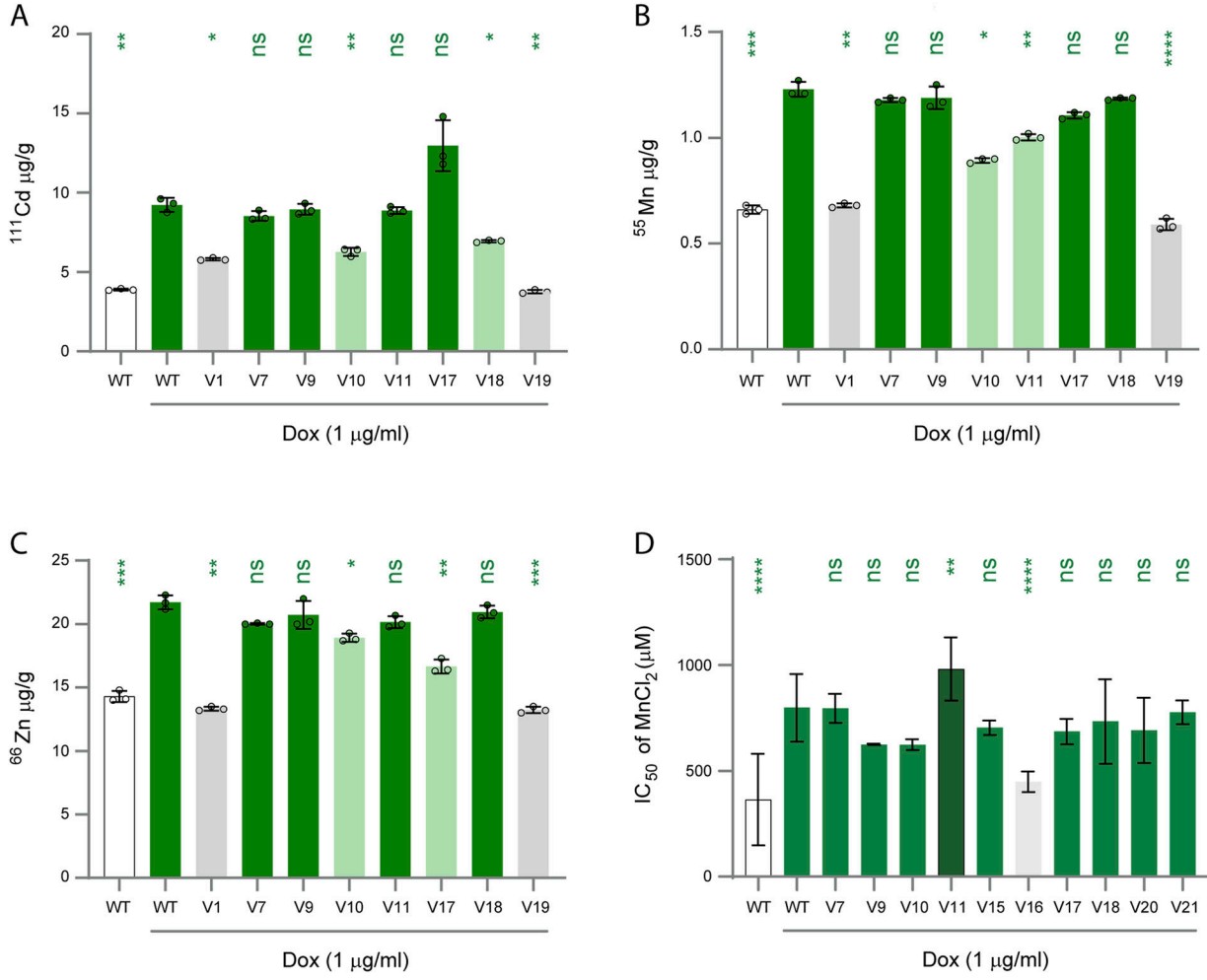

**Figure 3. ICP-MS and competitive transport between Zn$^{2+}$ and Mn$^{2+}$ of ZIP8 variants.**
(**A, B, C**) Levels of intracellular Cd$^{2+}$, Zn$^{2+}$, and Mn$^{2+}$ were measured with ICP-MS in HEK SLC39A8$^{WT}$ and HEK SLC39A8$^{V1-21}$ cells after 48 h of doxycycline induction and 24 h of incubation with 0.5 $\mu$M CdCl$_2$. Green indicates WT-like activity, light green indicates decreased activity, and grey indicates no activity. Three biological replicates were performed per variant. An ordinary one-way ANOVA test was used to determine statistical significance, and multiple comparisons were analysed against WT$_{+DOX}$ (green) (ns, not significant; *$P < 0.0332$; **$P < 0.0021$; ***$P < 0.0002$; ****$P < 0.0001$). (**D**) Zinc and manganese competitive transport in HEK SLC39A8$^{WT}$ and HEK SLC39A8$^{V1-21}$ cells. The concentration of MnCl$_2$ needed to inhibit 50% of zinc transport (IC$_{50}$) is graphed. Green indicates WT-like, dark green indicates higher than WT, and grey indicates less than WT. Three biological replicates were performed per variant. An ordinary one-way ANOVA test was used to determine statistical significance, and multiple comparisons were analysed against induced (green) ZIP8$^{WT}$ expression (ns, not significant; **$P < 0.0021$; ****$P < 0.0001$).

genetic variants present in an average genome and already existing in the human population ([51], [52]). It is particularly important to focus on genes associated with human diseases, especially those implicated at the interface between the human system and the environment as they could lead to beneficial interventions. The *hSLC39A8* gene serves as an example, encoding the SLC39A8 protein, commonly known as ZIP8, a membrane transporter responsible for the cellular uptake of essential divalent metal ions such as Cd$^{2+}$, Mn$^{2+}$, and Zn$^{2+}$. Our study investigated the expression, localization, cellular ion uptake, and dimerization properties of 21 *hSLC39A8* genetic variants to better understand their corresponding pathogenicity.

Upon comparing our findings with previously reported literature, we found that consistent with our results, V3 (Gly38Arg) and V6 (Cys113Ser) were both shown in HeLa cells to have high ER

expression and disrupted Mn$^{2+}$ uptake ([6]). Our in silico homodimer analysis revealed that Gly38Arg is in proximity to sites potentially involved in homodimerization, located in the extracellular domain at the dimerization interphase. Therefore, this mutation may destabilize dimerization and contribute to the observed variant effect on ER retention. A previous study in HeLa cells found WT-like uptake of Cd$^{2+}$ for both variants and decreased Se$^{4+}$ uptake for V3 but did not examine the impact of the mutations on localization ([12]). Although there is this discrepancy, V3 and V6 were both reported separately in multiple individuals with CDG type II and Leigh-like syndrome diseases, respectively, and the current clinical classification coincides with our classification of both variants as pathogenic ([Tables 1] and [2]). In addition, we examined V5, an arginine substitution at residue 113, that was reported in one individual with CDG type II but, because of the lack of strong evidence, was

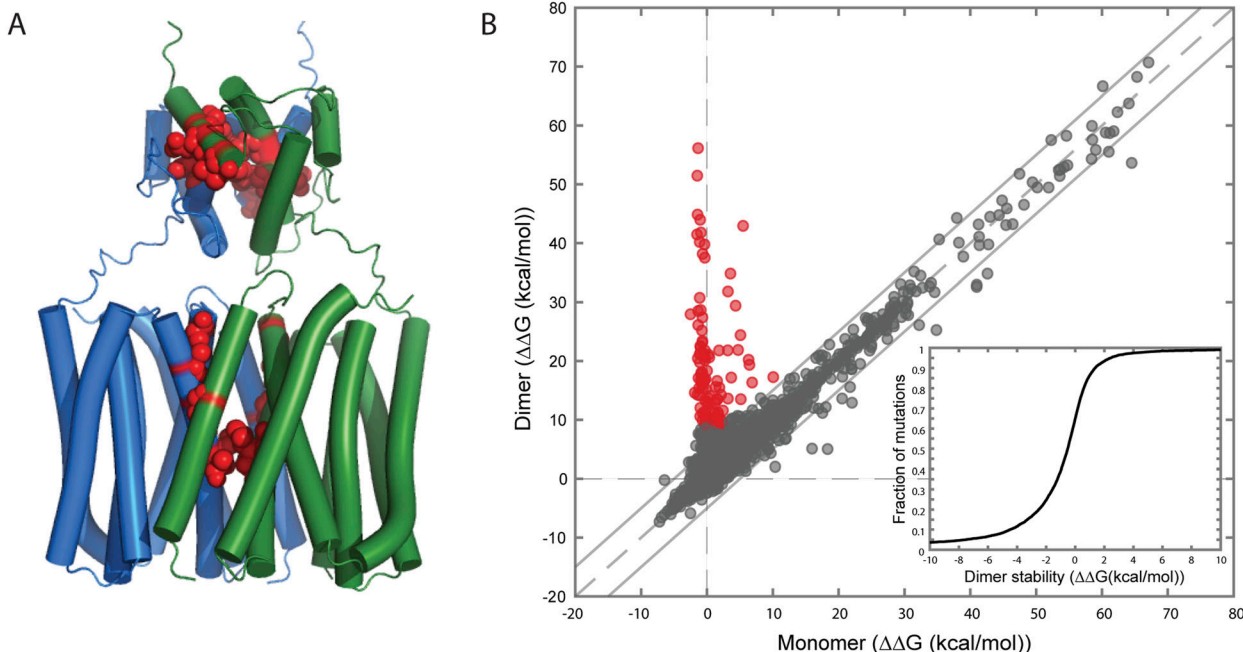

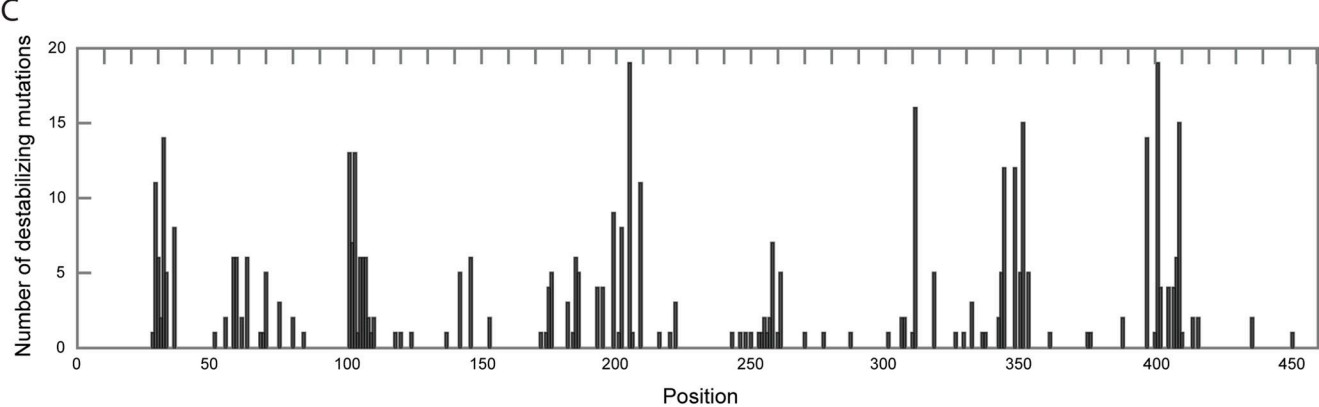

**Figure 4. Structure model of SLC39A8 dimer and mutations critical for dimer stability.**
**(A)** AlphaFold model of the SLC39A8 dimer. Monomers are shown in blue or green. Strongly destabilizing residues in the first percentile of the stability distribution are shown in red. **(B)** Comparison of thermodynamic changes caused by single mutations in the monomer versus the dimer **(inset)**. Distribution of single mutation effects on dimer stability, calculated as described in Equation (1). The upper fifth percentile contains 442 most destabilizing substitutions distributed over 105 positions. The first percentile contains 87 most destabilizing single amino acid substitutions distributed over 17 sites (highlighted in red). **(C)** Dimer-destabilizing mutations distribute in clusters. We plotted the upper fifth percentile of the most destabilizing mutations over the length of the protein detecting a heterogeneous distribution of variants per position.

classified as VUS. Our study reveals that this mutation is pathogenic because of diminished protein expression, leading to a lack of Cd$^{2+}$ uptake (Table 2). Indeed, the cysteine at residue 113 is highly conserved in mammals (18) and the importance of the cysteine at residue 113 is also illustrated by the AlphaMissense predictions, as both V5 and V6 were assigned highly pathogenic scores (Table 2).

Several *hSLC39A8* genetic variants co-occur in patients, and therefore, it is important to study the variants separately to distinguish their individual impact and contribution to disease manifestations. V2 (Val33Met) was initially discovered in a patient with two other variants, V9 (Gly204Cys) and V13 (Ser335Thr). A subsequent study looked only at characterizing the impact of all

three variants together on ZIP8 in HeLa cells (6, 21, 31). For the first time, we experimentally characterized these variants separately in cells to discern their individual impact on ZIP8. Whereas V2 was clinically uncategorized because of conflicting interpretations, we now provide experimental evidence that it represents a loss-of-function mutation caused by loss of protein expression, classifying it as a pathogenic variant (Table 2). Furthermore, according to our analysis, this variant is located within a region potentially involved in homodimerization. Therefore, this mutation may lead to dimer destabilization, which could contribute to the observed effect on protein expression. For V9 and V13, which were both clinically uncategorized because of conflicting interpretations and lack of

evidence, respectively, we now provide experimental evidence that V9 is benign, whereas V13 is pathogenic (Table 2). Our findings are consistent with another study showing V9 with WT-like $Cd^{2+}$ cellular accumulation and V13 with loss of $Cd^{2+}$ accumulation (12). V9 may therefore be a neutral passenger variant, and the cause of CDG type II in this patient may be attributed to the other two mutations Val33Met and Ser335Thr.

V4 (Ser44Trp) and V16 (Gly350Arg) were also discovered together in two siblings with SLC39A8-CDG and Leigh-like syndrome phenotypes and have been unclassified clinically (29). Our study provides evidence that both variants are pathogenic variants, with the Ser44Trp mutation causing ER retention and the Gly350Arg mutation causing disruption in ion uptake (Table 2). Supporting this, we observed that V16 falls within a potential cluster of residues vulnerable to mutations that destabilize dimerization and may prevent functional dimer formation. This could explain the lack of ion uptake despite proper expression and PM localization. V14 (Ile340Asn) was another variant discovered in co-occurrence with V3 (Gly38Arg) but has not yet been studied individually and is currently clinically uncategorized (6, 21, 31). We showed that the Ile340Asn substitution caused ER retention of the transporter and loss of cellular uptake of $Cd^{2+}$, and should therefore be classified as pathogenic (Table 2).

The highly pleiotropic and highly frequent (allele frequency of $4.47 \times 10^{-2}$) V19 (Ala391Thr) was discovered in a patient with SLC39A8-CDG and has been reported in numerous GWAS to be highly associated with various biological phenotypes and disorders (10, 16, 29, 33). The expression of the Ala391Thr variant in cells has produced conflicting results. In terms of expression levels, it has been shown in HEK293 cells to express to a similar level as WT (53) or to a lesser extent (54). In CHO cells (55) and MDCK cells (56), the expression and PM localization of V19 were not different from WT. Functionally, the Ala391Thr variant had decreased $Cd^{2+}$ uptake in HEK293 cells (53), a lower $Zn^{2+}$ uptake in CHO cells (55), and no difference in MDCK cells (56). In the latter study, the effect of the mutation was only revealed upon preincubation of MDCK cells with $Mn^{2+}$ and $Zn^{2+}$, as Ala391Thr prevented a manganese-dependent increase in PM levels of ZIP8 and $Zn^{2+}$-dependent reduction in PM levels of ZIP8 (13, 56). However, looking at the data, one could argue that these ions may affect the expression levels of ZIP8 as well (56). The effect of $Mn^{2+}$ or $Zn^{2+}$ preincubation and ZIP8 expression has not been investigated in cells other than MDCKs and could potentially be the underlying cause for the differences seen in expression or function.

Compounding this inconsistency, Ala391Thr was classified as benign according to ClinVar and predicted as benign by Alpha-Missense (Table 2). Prediction algorithms like AlphaMissense are trained based on a structural context and are limited in their ability to address more physiological properties such as the effect of a variant on interactions with other proteins. Unlike most studies, we found that the Ala391Thr substitution caused a loss of uptake of all ions tested because of impaired protein expression, eliciting the classification of this variant as pathogenic in our study (Table 2). The difference in our results to others may be due to the expression system. When we substituted the same alanine at residue 391 to a serine instead of a threonine (V20), the resulting substitution produced a ZIP8 transporter with proper PM localization and uptake

of $Cd^{2+}$, $Zn^{2+}$, and $Mn^{2+}$ in the cell (Table 2). Furthermore, the substitution of another nearby residue, Asn389, to a serine (V18) also resulted in proper expression and PM localization (Table 2). The pathogenic property of V19 is therefore unique and warrants further studies.

V7 (Asp193Asn) and V21 (Leu449Phe) were both included in our study because of their previous associations with non-pathogenic phenotypes, and our aim was to contextualize possible biological rationales for such associations. Specifically, the Asp193Asn substitution has been associated with changes in speech and psychomotor speed, whereas the Leu449Phe mutation correlates with changes in grey matter volume and alcohol consumption (57). The Asp193Asn substitution was predicted to be benign by Alpha-Missense, and our study presents the variant as benign as well because of the observed WT-like accumulation of both $Zn^{2+}$ and $Mn^{2+}$ with ICP-MS (Table 2). With our uptake assays, we saw a decreased uptake for both $Zn^{2+}$ and $Mn^{2+}$; this may be due to the differences in the experimental setup including higher concentrations of ions exposed and shorter duration of incubation time. The decreases seen with the uptake assays in V7 may be attributed to the variant location at the dimer interface, but whether this is associated with the correlating phenotype requires further investigation. The Leu449Phe variant is highly frequent (allele frequency of $1.08 \times 10^{-2}$), and consistent with the AlphaMissense scoring of benign, we observed WT-like localization and cellular uptake for all three ions tested. The distribution of trace elements in the brain is organized within distinct regions, with higher concentrations of iron, copper, zinc, and selenium in the grey matter regions (58). Therefore, a comprehensive characterization of the transport activity for all ZIP8 substrates would be interesting, as the findings could potentially elucidate the correlated phenotypes of Leu449Phe.

The remainder of the variants, V8 (Phe203Ser), V10 (Trp306Cys), V11 (Thr308Met), V12 (Ile322Thr), V15 (Ile340Met), and V17 (Gln364Arg), are all classified as VUS or uncharacterized because of conflicting evidence (Table 1). Other than V12 (Ile322Thr) that affected the PM localization of ZIP8 and is pathogenic, the other variants were all detectable on the PM (Table 2). V15 demonstrated WT-like cellular uptake of all the ions tested and is benign in our study (Table 2). V8 (Phe203Ser) lacked cellular uptake for all tested substrates, and consistent with AlphaMissense, it is pathogenic in our study (Table 2). Given that the Phe203 residue is located in a region potentially crucial for homodimer formation, the Phe203Ser substitution may disrupt proper dimerization, thereby contributing to its pathogenicity. V10 (Trp306Cys) exhibited decreased cellular uptake of $Mn^{2+}$ and decreased measures of accumulated $Cd^{2+}$, $Zn^{2+}$, and $Mn^{2+}$, and was therefore, in accordance with AlphaMissense, likely pathogenic within our study (Table 2). V11 (Thr308Met) exhibited WT-like uptake of $Cd^{2+}$ and $Zn^{2+}$ but diminished $Mn^{2+}$ cellular uptake (Table 2). Further validation through a competition assay revealed a higher $IC_{50}$ of $Mn^{2+}$ is required to compete with $Zn^{2+}$ transport, compared with $ZIP8^{WT}$, and is likely pathogenic within our study (Fig 3, Table 2). Both the Trp306 and Thr308 residues are in proximity to potential sites involved in homodimer formation, suggesting that the variant phenotypes may be for this reason. In addition, Thr308 is adjacent to Cys310, which, when mutated to alanine, has been shown to increase the affinity for $Zn^{2+}$ over other

ion substrates ([34]). The phenotypic consequences of this substitution and its impact on affinity, as well as its association with disease, remain to be elucidated in the human population. Lastly, V17 (Gln364Arg) showed WT-like uptake but decreased accumulated levels of $Zn^{2+}$, and decreased uptake but WT-like accumulated levels of $Mn^{2+}$ and is therefore likely pathogenic within this study ([Table 2]).

The use of an overexpression system with cell-based assays to study transporters bypasses the need to use highly specialized techniques, such as patch clamp, radiolabelled isotopes, or reconstituted liposomes. However, using these assays to infer transport activity has its limits. Specifically, within this study, we are limited by the affinity and kinetics of the dyes used (e.g., FLIPR-5, Zinpyr-1), and the long duration of certain experiments (e.g., cell toxicity, ICP-MS), and do not translate into direct transport function because of other cellular complexities. We therefore combined a number of different experimental approaches with computational analysis and VEPs in this study. Ultimately, we characterized 21 variants in the protein-coding region of the *hSLC39A8* gene, encoding SLC39A8 (ZIP8), and as a result, we now suggest new clinical classifications for 16 variants with uncategorized or VUS clinical status. By employing both experimental and computational approaches, we display the possibility of investigating the functional consequence of poorly characterized genetic variants within human transporter genes. As more analyses such as ours are performed, we expect improvements in the variant effect prediction algorithms to better serve clinical interpretations of disease-causing mutations.

# Materials and Methods

### Variant selection

Variants were collated based on two approaches: a review of the literature, and a programmatic mining of publicly available datasets to curate a collection of variants with annotated clinical and biological impacts. Datasets were obtained from ClinVar (https://www.ncbi.nlm.nih.gov/clinvar) ([59]), data freeze 01.02.2022, Genebass (https://app.genebass.org/, version 0.7.8-alpha) ([60]), Open Targets Genetics (https://www.opentargets.org/, version 6) ([61]), the IEU Open GWAS Project collection of GWAS (https://gwas.mrcieu.ac.uk/, version 5.9.0) ([62] *Preprint*), and LitVar (https://www.ncbi.nlm.nih.gov/research/litvar2/) ([63]). The mutation nomenclature used in this study was based on the coding region of the solute carrier family 39, member 8 (SLC39A8) transcript (ENSG00000138821), and GenBank Accession NM_001135146.2 (cDNA), following the recommendations of the Human Genome Variation Society (HGVS; https://www.hgvs.org/mutnomen); and the cDNA numbering of nucleotide variants, according to journal guidelines (https://www.hgvs.org/mutnomen).

Using these approaches and other considerations (in silico prediction of thermodynamic stability and variant effect, inclusion of negative and positive controls, and clinical significance), we selected 21 variants within the coding region of the *hSLC39A8* gene. We prioritized variants linked to interesting phenotypes, predicted to be highly destabilizing, classified clinically as

uncertain significance (VUS) or uncategorized, and highly frequent in the human population. For convenience, variants were annotated numerically from the N to C terminus ([Table 1]). All variants are summarized in [Table 1], and if applicable, we include the minor allele frequency obtained from the Genome Aggregation Database (gnomAD) across all populations (https://gnomad.broadinstitute.org, version 3.1.2) ([64] *Preprint*) and the Regeneron Genetics Center (RGC) Million Exome data (https://www.regeneron.com/science/genetics-center) ([65] *Preprint*).

### Structure prediction and identification of key residues for multimerization

We used AlphaFold2 to explore the folding of WT SLC39A8/ZIP8 protein ([66]). Because multimerization has been described for many members of the ZIP family ([67], [68], [69], [70]), we studied ZIP8 as a dimer and trimer, using AlphaFold multimer ([71] *Preprint*). To assess model quality, we used the per-residue local distance difference test (pLDDT), reported by AlphaFold. To identify residues essential for multimerization, we used FoldX ([72]) to estimate changes in thermodynamic stability caused by a mutation in the monomeric ($\Delta\Delta G_m(i)$) and dimeric ($\Delta\Delta G_d(i)$) forms. To identify destabilizing mutations, we estimated the changes in stability because of dimerization for each possible mutation as follows:

$$Dimer\ stability = \Delta\Delta G_d(i) - 2\Delta\Delta G_m(i) = \Delta G_d(i) - 2\Delta G_m(i) - (\Delta G_d(wt) - 2\Delta G_m(wt)). \tag{1}$$

Mutations causing significant changes in the stability of the monomer compared with the dimer were identified as the upper fifth percentile of the distribution of stabilities.

### VEPs

To assess the effect of missense variants on the WT ZIP8, we used AlphaMissense ([73]), a state-of-the-art prediction method based on deep learning. This method uses both sequence and structure information to classify variants based on a continuous score ranging from 0.0 to 1.0. Pathogenic variants (P) have scores ranging between 0.56 and 1.00; benign variants (B) have scores ranging between 0.0 and 0.34; and otherwise, variants are classified as ambiguous or uncertain (U). We used the precalculated scores reported by Reference ([73]).

### Cell line generation

HEK293 cell lines stably overexpressing the WT or variant human *SLC39A8* (*hSLC39A8*) genes were generated in a doxycycline-inducible manner using the Jump In T-REx HEK293 system (#A15008; Thermo Fisher Scientific) as previously described ([74]). These cells are referred to as HEK SLC39A8[WT] and HEK SLC39A8[V1-V21] throughout the article. The *hSLC39A8* gene harbouring the WT or variant sequences was codon-optimized and carries C-terminal STREPII and HA tags (#132177; Addgene) (generated by GenScript). The WT or the variant *hSLC39A8* gene was cloned into a JTI R4 CMV-TO MCS pA vector backbone (#A15004; Thermo Fisher Scientific) and introduced into the Jump In parental cell line using Lipofectamine

3000 Transfection Reagent (#L3000001; Thermo Fisher Scientific) according to the manufacturer's protocol. The transfected cells were selected over a 14-d period with G418 Geneticin (A1720-5G; Sigma-Aldrich) and then tested for mycoplasma contamination. All cells were cultured in DMEM (#D5796; Sigma-Aldrich) supplemented with 10% FCS (#S1810-500; BioWest), with 1% penicillin and 1% streptomycin (#P4333; Sigma-Aldrich) at 37°C with 5% $CO_2$ and 95% humidity.

### Immunofluorescence imaging and analysis

HEK SLC39A8[WT] and HEK SLC39A8[V1-V21] cells were seeded (at $1 \times 10^4$ cells/well) into 96-well plates previously coated with poly-L-lysine for 16 h and induced for protein expression with doxycycline (1 µg/ml) for 24 h. For ZIP8 staining against the PM marker CellBrite Red (#30023; Biotium), cells were fixed with 4% formaldehyde in PBS (#D8537; Sigma-Aldrich) for 10 min at RT. For ZIP8 staining against the endoplasmic reticulum (ER) marker calreticulin, cells were fixed with ice-cold methanol for 10 min at –20°C and rehydrated with PBS for 10 min at RT. Cells were then blocked with blocking buffer (0.5% BSA and 0.15% glycine in PBS) for 1 h at RT, and primary antibody staining was performed overnight at 4°C. The following primary antibodies were used and diluted in blocking buffer: anti-HA (rat) (#47877600; Roche) at 1:1,000 dilution and anti-calreticulin (rabbit) (#ab92516; Abcam) at 1:250 dilution. The next day, cells were washed with blocking buffer and stained with secondary antibody for 1 h at RT. The following secondary antibodies were used and diluted at 1:500 in blocking buffer: Alexa Fluor 488 goat anti-rat (#A-11006; Thermo Fisher Scientific) and Alexa Fluor 568 (#A-11011; Thermo Fisher Scientific) goat anti-rabbit (Invitrogen). Cells were then washed with blocking buffer, and stained with DAPI (#D9542; Sigma-Aldrich) and, if applicable, CellBrite Red according to the manufacturer's protocol. Lastly, cells were washed with PBS before imaging on the Opera Phenix (PerkinElmer) high-throughput microscope (40x magnification). Image processing and analysis were performed as previously described (74). Briefly, 4–20 images per variant cell line and 81–135 images for the WT cell line were processed and analysed using ImageJ. Pearson's correlation coefficient for the PM ($r_{PM}$) or the ER ($r_{ER}$) was calculated per WT or variant image. The membranes were immunoblotted with mouse anti-HA (H6533; Sigma-Aldrich) and mouse anti-GAPDH (sc-365062; Santa Cruz). Bound antibodies were detected with horseradish peroxidase–conjugated secondary goat anti-mouse antibodies using ECL (Thermo Fisher Scientific).

### Immunoblot analysis

HEK SLC39A8[WT] and HEK SLC39A8[V1-V21] cells were seeded in six-well plates and grown to confluency before harvesting for sample preparation. ZIP8 expression was induced with doxycycline (1 µg/ml) for 24 h before harvest. Cells were washed with PBS before harvesting with a Nonidet P-40 lysis buffer (50 mM Hepes, pH 7.4, 250 mM NaCl, 5 mM EDTA, 1% NP-40, 1x Roche EDTA-free protease inhibitor cocktail [Sigma-Aldrich]) for 30 min on ice. Lysates were precleared with centrifugation (16,000g, 15 min, 4°C), and proteins were quantified with the Bradford assay (Bio-Rad), according to the manufacturer's protocol.

### Cd[2+] uptake using the FLIPR Calcium-5 assay

For the measurement of Cd[2+] uptake by HEK SLC39A8[WT] and HEK SLC39A8[V1-V21] cells, we used the FLIPR Calcium-5 Assay kit (#R8185; Molecular Devices) as previously described (36). Briefly, cells were seeded (at $1 \times 10^4$ cells/well) into 96-well plates previously coated with poly-L-lysine for 24 h, and then, if applicable, protein expression was induced with doxycycline (2 µg/ml) for 24 h. The next day, cellular media were exchanged with the FLIPR Calcium-5 dye, previously prepared according to the manufacturer's protocol, for a 30-min incubation at 37°C with 5% $CO_2$. The fluorescence signal ($\lambda_{Ex}$ = 470–495 nm; $\lambda_{Em}$ = 515–575 nm) was then recorded with the SpectraMax i3X plate reader (Molecular Devices).

For the measurement of Cd[2+] uptake, fluorescent signals were recorded before substrate addition at two reads per minute for 5 min. The recording was then paused for cadmium chloride (CdCl$_2$) substrate addition and resumed at two reads per minute for 30 min (Fig 2A). To quantify Cd[2+] uptake for each concentration of CdCl$_2$ addition, the recordings were normalized to the measurements before substrate addition and the area under the curve (AUC) from 10 to 30 min post-substrate addition was calculated using the trapezoidal rule (Fig 2A). For CdCl$_2$ titration, the calculated AUC for Cd[2+] uptake was then plotted against the concentration of added CdCl$_2$ and an $EC_{50}$ of 2.64 µM (95% Cl: 2.13, 3.44) for ZIP8[WT] was determined based on a non-linear regression curve fitting method (GraphPad). For determining Cd[2+] uptake for each WT or variant ZIP8, the AUC for Cd[2+] uptake was measured and calculated in each cell line with a set CdCl$_2$ concentration of 2 µM (Fig 2B).

### Zn[2+] uptake using Zinpyr-1

To measure Zn[2+] uptake, we used the intracellular Zn[2+]-sensitive dye Zinpyr-1 as previously described (38, 39). Briefly, HEK SLC39A8[WT] and HEK SLC39A8[V1-V21] cells were seeded in a six-well plate (at $6 \times 10^5$ cells/well) for 24 h and protein expression was induced with doxycycline (1 µg/ml) for 24 h. The next day, cells were incubated with 50 µM of Zinpyr-1 in experimental buffer consisting of HBSS (#14175095; Gibco) and 25 mM Hepes (#15630056; Gibco) for 30 min at 37°C. Cells were then harvested by trypsin (#T3924; Sigma-Aldrich) and incubated with different concentrations of zinc chloride (ZnCl$_2$) in experimental buffer for 10 min at 37°C before washing and resuspending the cells in flow cytometry buffer (10% FBS in PBS) for flow cytometry analysis. Fluorescence was measured using BD LSRFortessa Cell Analyzer ($\lambda_{Ex}$ = 492 nm, $\lambda_{Em}$ = 527 nm) and analysed with FlowJo software.

### Zn[2+] and Mn[2+] competition assay

For the Zn[2+] and Mn[2+] competition assay, cells were plated (at $1 \times 10^4$ cells/well) in a 96-well plate format previously coated with poly-L-lysine for 24 h, and then, if applicable, protein expression was induced with doxycycline (1 µg/ml) for 24 h. The next day, cells were incubated with Zinpyr-1 as described above. Cells were then washed with experimental buffer and co-incubated with ZnCl$_2$ (40 µM) and MnCl$_2$ (0 to 2.56 mM) for 10 min at 37°C. Zinpyr-1 fluorescence was measured with Opera Phenix (PerkinElmer) ($\lambda_{Ex}$ = 492 nm, $\lambda_{Em}$ = 527 nm), and the images were analysed with ImageJ.

### ICP-MS

Cells were plated on 15-cm plates and grown to confluency before harvesting for sample preparation. ZIP8 expression was induced with doxycycline (1 μg/ml) for 48 h before harvest, and cells were incubated with 0.5 μM $CdCl_2$ for 24 h before harvest. Cells were harvested with trypsin and washed with PBS before pellets were weighed and frozen with liquid nitrogen. Samples were then sent to BOKU Core Facility Mass Spectrometry for analysis.

Chemicals used at the facility: Water for solution preparation was purified employing an Ultra Clear basic reverse osmosis system (SG Wasseraufbereitung und Regenerierstation GmbH) and subsequently sub-boiled (MLS, Leutkirch). Nitric acid (p.a.), hydrogen peroxide 30% (Suprapur), and a certified multi-element standard solution VI were purchased from Merck (KGaA). Nitric acid was double-sub-boiled and stored in a perfluoroalkoxy-polymer bottle.

Instrumentation used: A MW7000 from Anton Paar was used for microwave-assisted acid digestion with nitric acid/hydrogen peroxide. An iCAP quadrupole ICP-MS (Thermo Fisher Scientific) was used for elemental analysis. The system was operated in kinetic energy discrimination mode; that is, the reaction/collision cell was filled with helium (4.8 ml $min^{-1}$). Cadmium was detected at m/z 111, and zinc was detected at m/z 66. Multi-element standards were employed to calibrate the system for quantification.

Sample preparation and experimental setup: All materials that came into contact with the samples, blanks, and standards were precleaned by soaking for 24 h in 10% and 24 h in 1% nitric acid followed by flushing with ultra-pure water and drying under a class 100 laminar flow bench. One replicate (n = 1) per sample was digested and measured. Each sample for digestion is around 50–100 mg and digested with 500 μl nitric acid and 30 μl $H_2O_2$. Nitric acid blanks (2% [vol/vol]) were measured between the sample runs for instrumental blank control.

Quantification was performed using multilevel external calibration with multi-element standards and internal standardization via indium.

### $Mn^{2+}$ uptake using $Mn^{2+}$ cell toxicity

For quantification of $Mn^{2+}$ uptake, we used the cell death properties of $Mn^{2+}$ observed in HEK293 cells with an overexpression of the ZIP8 transporter (Fig 2E). HEK SLC39A8[WT] and HEK SLC39A8[V1-V21] cells were seeded (at 1 × 10⁴ cells/well) into 96-well plates previously coated with poly-L-lysine for 24 h, and then, if applicable, protein expression was induced with doxycycline (1 μg/ml) for 24 h. Cells were then incubated with differing concentrations of manganese chloride ($MnCl_2$) in culture media for 48 h, and cellular viability was then assessed with CellTiter-Glo Luminescent Cell Viability Assay (#G7570; Promega) according to the manufacturer's protocol. The luminescence was measured with the SpectraMax i3X plate reader (Molecular Devices).

### Ion uptake classification

For each ion uptake assay, we characterized the cellular uptake for each ion and classified each variant into WT, decreased, or no response. This was based on the comparison of uptake in HEK SLC39A8[V1-V21] cells with the uptake in either doxycycline-induced (for ZIP8[WT] activity) or uninduced (for the absence of ZIP8[WT] activity) HEK SLC39A8[WT] cells. The WT category is defined as when the variant-expressing cell shows significantly higher uptake than the uninduced HEK SLC39A8[WT] cells and not significantly different than induced HEK SLC39A8[WT] cells. The decreased category is defined as when the variant-expressing cell shows significantly lower uptake than the induced HEK SLC39A8[WT] cells and significantly higher uptake than the uninduced HEK SLC39A8[WT] cells. As well, the decreased category is also true for when the variant-expressing cell shows no statistical difference in uptake with the induced HEK SLC39A8[WT] cells nor the uninduced HEK SLC39A8[WT] cells. Lastly, the no response category is defined as when the variant-expressing cell shows significantly lower uptake than the induced HEK SLC39A8[WT] cells and not significantly different than uninduced HEK SLC39A8[WT] cells.

## Supplementary Information

## Acknowledgements

We would like to thank the laboratory of Giulio Superti-Furga for all the fruitful discussions and Andrè Ferdigg for the original idea to use $Mn^{2+}$ toxicity as a transport assay. We thank Dr. Chan Xiong and Ing. Andreas Brandstätter for conducting MS experiments. The MS equipment was kindly provided by the BOKU Core Facility Mass Spectrometry. This work was partly performed by the RESOLUTE (https://re-solute.eu/) and REsolution (https://re-solute.eu/resolution) consortia. Plasmids are available through Addgene (https://www.addgene.org/depositor-collections/re-solute/). RESOLUTE has received funding from the Innovative Medicines Initiative 2 Joint Undertaking under grant agreement No 777372. This Joint Undertaking receives support from the European Union's Horizon 2020 research and innovation programme and EFPIA. REsolution has received funding from the Innovative Medicines Initiative 2 Joint Undertaking under grant agreement No 101034439. This Joint Undertaking receives support from the European Union's Horizon 2020 research and innovation programme and EFPIA. Continuation of the work after the funding period occurred via the Austrian Academy of Sciences. This article reflects only the authors' views, and neither IMI nor the European Union and EFPIA are responsible for any use that may be made of the information contained therein.

### Author Contributions

W-A Wang: conceptualization, data curation, formal analysis, validation, investigation, visualization, methodology, and writing—original draft, review, and editing.

A Garofoli: data curation, investigation, and writing—original draft, review, and editing.

E Ferrada: data curation, software, formal analysis, validation, investigation, and writing—review and editing.

C Klimek: validation, investigation, and methodology.

B Steurer: conceptualization, investigation, visualization, methodology, and writing—review and editing.

A Ingles-Prieto: conceptualization and writing—review and editing.

T Osthushenrich: data curation, investigation, methodology, and writing—review and editing.

A MacNamara: conceptualization and supervision.

A Malarstig: conceptualization and supervision.

T Wiedmer: conceptualization, supervision, project administration, and writing—review and editing.

G Superti-Furga: conceptualization, supervision, funding acquisition, and writing—review and editing.

**Conflict of Interest Statement**

G Superti-Furga is the scientific founder and a shareholder of Proxygen and Solgate, the latter a SLC-focused company. The rest of the authors declare that they have no known competing financial interests or personal relationships that could have appeared to influence the work reported in this study.

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
