## [Reviewer comments · Life Science Alliance]

Life Science Alliance

Human genetic variants in SLC39A8 impact uptake and steady-state metal levels within the cell.

Wen-An Wang, Andrea Gaorfoli, Evandro Ferrada, Christoph Klimek, Barbara Steurer, Alvaro Ingles-Prieto, Tanja Osthusenrich, Aidan MacNamara, Anders Malarstig, Tabea Wiedmer, and Giulio Superti-Furga

DOI: <https://doi.org/10.26508/lsa.202403028>

Corresponding author(s): Giulio Superti-Furga, CeMM Research Center for Molecular Medicine and Wen-An Wang, CeMM Research Center for Molecular Medicine

Review Timeline:

Submission Date:	2024-09-02
Editorial Decision:	2024-10-16
Revision Received:	2024-12-23
Editorial Decision:	2025-01-06
Revision Received:	2025-01-15
Accepted:	2025-01-15

Transaction Report:

October 16, 2024

Re: Life Science Alliance manuscript #LSA-2024-03028

Prof. Giulio Superti-Furga
CeMM Research Center for Molecular Medicine of the Austrian Academy of Sciences
Lazarettgasse 14
Vienna 1090
Austria

Dear Dr. Superti-Furga,

Thank you for submitting your manuscript entitled "Human genetic variants in SLC39A8 impact metal transport specificity" to Life Science Alliance. The manuscript was assessed by expert reviewers, whose comments are appended to this letter. We invite you to submit a revised manuscript addressing the Reviewer comments.

Please note that the suggestions Reviewer 3 refers to are included on the marked-up pdf attached to this email.

Thank you for this interesting contribution to Life Science Alliance. We are looking forward to receiving your revised manuscript.

Sincerely,

B. MANUSCRIPT ORGANIZATION AND FORMATTING:

Reviewer #1 (Comments to the Authors (Required)):

The paper of Wen-An Wang et al., investigates the effects of SLC3918 human genetic variants on SLC3918 subcellular localization and differential metal transport specificities. This study identified different variants with complete loss of transport function and five with differential transport activity between metal ions. This is an interesting study essential in the field to further understand the impact of these mutations on the diverse spectrum of human diseases. Most of the experiments testing the transport activity are indirect making difficult a robust conclusion. Please find the following comments to make a revised version of the paper acceptable.

1_ I don't understand the results shown in Fig1C. How can you have a correlation when no expression is observed for some variants? About V8, does the result means that this form is found both in the ER and at the plasma membrane? This maybe could maybe be accompanied with enlarged pictures showing for each group an example of ER localization, plasma membrane or both.

2_ The observed impacts of the different variants on the transport of Cd²⁺, Zn²⁺ and Mn²⁺ is difficult to appreciate without a precise quantification of SLC39A8 expression according to the cell lines stably overexpressing the WT or variant forms. This can be done easily by a western blot approach and the results on Cd²⁺/ Zn²⁺ and Mn²⁺ reanalyzed accordingly, if strong changes are observed in expression.

2_ The Zn transport experiment is difficult to understand. The supplementation of Zn has no effects in Wt cells somehow meaning that the cells arrive to correctly extrude the Zn excess. If you get a Zn excess by overexpressing SLC39A8, would that mean that SLC39A8 can be involved in the exit of Zn? How can you differentiate the intake from the exit in your experiments? This is very important to conclude about the function of SLC39A8.

3_ The Mn transport was measured by the percent survival (%) of HEK SLC39A8WT cells, induced and non-induced with doxycycline, in response to MnCl₂ (0-1 mM) treatment for 48 hours. 48 hours appears quite long to measure a transport activity. The results certainly mostly reflect indirect mechanisms capable to extrude manganese or not. Manganese can quench the Ca²⁺ sensor Fura-2 very efficiently and rapidly. Such experiments could confirm the results obtained after 48h.

General comment: I understand that such experiments are tricky but the authors could get a very important confirmation of their main results by using an ICP-MS approach available in most of the hospitals. This could strengthen the conclusions.

Reviewer #2 (Comments to the Authors (Required)):

This manuscript characterizes 21 human SLC39A8 (hSLC39A8) gene variants and evaluates their impact on protein localization and metal transport for Cd²⁺, Zn²⁺, and Mn²⁺. This study aims to contribute to the understanding of how hSLC39A8 variants affect ZIP8 function and their potential role in diseases related to metal ion dysregulation.

- Validation of doxycycline-induced expression: The immunofluorescence data presented for WT and variant hSLC39A8 shows protein localization but does not sufficiently validate the expression induced by doxycycline at the gene or protein level. qPCR and Western blot are essential to confirm that both WT and variant hSLC39A8 are expressed comparably after doxycycline induction. These experiments would ensure that any functional differences observed are not due to varying expression levels.

- Metal ion transport assays: The authors claim to have measured metal transport for Cd²⁺, Zn²⁺, and Mn²⁺ using the FLIPR Calcium-5 Assay (for Cd²⁺), Zinpyr-1 dye (for Zn²⁺), and a cell viability assay (for Mn²⁺). However, none of these assays directly measure metal transport. The FLIPR Calcium-5 Assay measures calcium flux, not Cd²⁺ transport. Similarly, Zinpyr-1 measures Zn²⁺ levels in the cytosol but not Zn²⁺ transporter activity, and cell viability assays reflect general cell health, not Mn²⁺ transport. To accurately assess metal transport, the authors should employ specific uptake assays, such as radiolabeled or fluorescently tagged metal ions. Alternatively, the manuscript should be revised to clearly state that these methods assess changes in ion

levels or viability, rather than direct transport activity.

- Conflicting reports on Ala391Thr variant expression: The manuscript cites conflicting reports regarding the expression of the Ala391Thr variant, but the current immunofluorescence data show no detectable protein expression of this variant in HEK 293 cells. This raises concerns about attributing the reduced Cd²⁺ and Zn²⁺ transport to functional defects when the lack of detectable expression could be the underlying cause. The authors should reconsider their interpretation and clarify that the observed functional deficits may result from insufficient protein levels rather than inherent defects in the transporter. Further validation of expression levels is needed to resolve this issue.
- Figure 1A (Line 105): The variant number should be indicated in the topology diagram, and the location of the HA tag must be clearly shown.
- Supplementary Figure 1B (Line 113): The authors should include Supplementary Figure 1B in the main figures or at least show images of variants with different cellular localizations, such as those in the endoplasmic reticulum (ER). Showing only the WT localization is insufficient, as its plasma membrane localization is already established.
- Immunofluorescence for variants (Line 119): The authors state that variants V2, V5, and V19 affect ZIP8 expression, yet the immunofluorescence images (Supplementary Figure 1) show no detectable protein expression for these variants. The authors should address this discrepancy.

Reviewer #3 (Comments to the Authors (Required)):

It is always a pleasure to review a manuscript that is exceptionally well-written and the data are unequivocal and on the cutting-edge of the field. Authors' abstract: The human SLC39A8 (hSLC39A8) gene encodes a plasma membrane protein SLC39A8 (ZIP8) that mediates the specific uptake of the metals Cd²⁺, Mn²⁺, Zn²⁺, Fe²⁺, Co²⁺, and Se⁴⁺. Pathogenic variants within hSLC39A8 are associated with congenital disorder of glycosylation type 2 (CDG type II) or Leigh-like syndrome. However, numerous mutations of uncertain significance are also linked to different conditions or benign traits. Our study characterized 21 hSLC39A8 variants and measured their impact on protein localization and transport for Cd²⁺, Zn²⁺, and Mn²⁺. We identified three variants, including the highly pleiotropic Ala391Thr, with disrupted protein expression, five variants with high retention in the endoplasmic reticulum and 12 variants with localization to the plasma membrane. From the 12 variants with plasma membrane localization, we identified three with complete loss of transport function, and five with differential transport activity between metal ions. Further in silico analysis on protein stability identified variants that may affect the stability of homodimer interfaces. This study elucidates the variety of effects of hSLC39A8 variants on ZIP8 and on diseases involving disrupted metal ion homeostasis.

This Reviewer has offered various suggestions in the text of the pdf file - which are mostly trivial but which is grammatically more correct and/or less confusing to the General Reader. The biggest concern is trivial: with articles involving metals and/or organic compounds that participate in oxidation-reduction, it is clearer to the Reader to avoid "reduces" or "reduction" when synonyms such as "lowered", "decreases", "diminution" can be used instead.

Other than that, I wholeheartedly recommend publication of this article with the highest priority. The study is a masterpiece in its field.

Dear Dr Sawey,

Please see below for a point-by-point response, in bold, to reviewer comments. We were able to perform qPCR and immunoblot analysis to show the incorporation and expression of wild-type and variant SLC39A8. We also selected a few variants to perform ICP-MS to confirm our findings. The results from these experiments have been incorporated into the manuscript. As well all the suggested changes to the text have been made in response to the reviewer's comments and suggestions. Thank you and we are happy to answer any questions. Looking forward to your response.

Reviewer #1 (Comments to the Authors (Required)):

The paper of Wen-An Wang et al., investigates the effects of SLC3918 human genetic variants on SLC3918 subcellular localization and differential metal transport specificities. This study identified different variants with complete loss of transport function and five with differential transport activity between metal ions. This is an interesting study essential in the field to further understand the impact of these mutations on the diverse spectrum of human diseases. Most of the experiments testing the transport activity are indirect making difficult a robust conclusion. Please find the following comments to make a revised version of the paper acceptable.

1_ I don't understand the results shown in Fig1C. How can you have a correlation when no expression is observed for some variants? About V8, does the result means that this form is found both in the ER and at the plasma membrane? This maybe could maybe be accompanied with enlarged pictures showing for each group an example of ER localization, plasma membrane or both.

We thank this reviewer for bringing this point to our attention. We used Pearson's correlation to measure the extent of the overlap between SLC39A8 and the PM or ER markers. This only takes into account the correlation and not the intensity of either stain. Therefore, one can only look at the relative positions of the clusters in Fig 1C to each other, as the images were taken under the same conditions. This rules out any background signals that may contribute to the high correlation values. We have now included enlarged sample pictures of each conditions/clusters in Figure 1C.

2_ The observed impacts of the different variants on the transport of Cd²⁺, Zn²⁺ and Mn²⁺ is difficult to appreciate without a precise quantification of SLC39A8 expression according to the cell lines stably overexpressing the WT or variant forms. This can be done easily by a western blot approach and the results on Cd²⁺/ Zn²⁺ and Mn²⁺ reanalyzed accordingly, if strong changes are observed in expression.

We thank this reviewer for making this point. We have now provided immunoblot analysis for all the cell lines quantifying SLC39A8 and GAPDH in Supplementary Figure 2. The western blot analysis confirms our immunofluorescence results and for all the variants that have functional transport, the expression levels of both the monomer band (~75kDa) and

the dimer band (~250kDa) are comparable to that of WT. For the variants that affect expression, we do not see HA positive bands and for variants that result in high ER retention, we do not see monomer and dimer bands but bands that are smaller in size suggesting these variants do not result in proper dimer conformations as WT. In general, if there are any differences observed, we do not think we can normalize the transport data to the SLC39A8 expression in whole lysate proteins, as we cannot know the level of active transporter on the plasma membrane in each cell line. We are now careful within the text to specify that what we see is not an effect on the transport function but the consequence on cellular uptake of the ion measured, this is done throughout the text in the result and discussion section.

2_ The Zn transport experiment is difficult to understand. The supplementation of Zn has no effects in Wt cells somehow meaning that the cells arrive to correctly extrude the Zn excess. If you get a Zn excess by overexpressing SLC39A8, would that mean that SLC39A8 can be involved in the exit of Zn? How can you differentiate the intake from the exit in your experiments? This is very important to conclude about the function of SLC39A8.

We appreciate the point of SLC39A8 expression and increased intracellular Zn being a factor that influences the activity of Zn exporters, as this could be a possibility that influences the claims of our results. We have now made changes accordingly in the manuscript, in the results and discussion section, to indicate that what we see is changes in ion levels within the cell and not a direct implication on transport function. We also include this as a limitation in the discussion for cell-based assays of this nature (line 322-327).

3_ The Mn transport was measured by the percent survival (%) of HEK SLC39A8WT cells, induced and non-induced with doxycycline, in response to MnCl₂ (0-1 mM) treatment for 48 hours. 48 hours appears quite long to measure a transport activity. The results certainly mostly reflect indirect mechanisms capable to extrude manganese or not. Manganese can quench the Ca²⁺ sensor Fura-2 very efficiently and rapidly. Such experiments could confirm the results obtained after 48h.

We appreciate the point of SLC39A8 expression and increased intracellular Mn being a factor that influences the activity of Mn exporters, as this could be a possibility that influences the claims of our results. We have now made changes accordingly in the manuscript to indicate that what we see is changes in ion levels within the cell and not a direct implication on transport function. We also include this as a limitation in the discussion for cell-based assays of this nature (line 322-327). We had previously performed Fura-2 experiments and produced similar results, we did not include this as the variability for these experiments were very high to perform accurate statistics.

General comment: I understand that such experiments are tricky but the authors could get a very important confirmation of their main results by using an ICP-MS approach available in most of the hospitals. This could strengthen the conclusions.

We thank the reviewer for making this suggestion to strengthen the conclusions of our paper. We now include ICP-MS data in Figure 3 to contribute to the validation of the results of all our assays, the experimental data are described in the result section (line 167-174) and throughout the discussion section. The classifications are also included in table 2.

Reviewer #2 (Comments to the Authors (Required)):

This manuscript characterizes 21 human SLC39A8 (hSLC39A8) gene variants and evaluates their impact on protein localization and metal transport for Cd²⁺, Zn²⁺, and Mn²⁺. This study aims to contribute to the understanding of how hSLC39A8 variants affect ZIP8 function and their potential role in diseases related to metal ion dysregulation.

- Validation of doxycycline-induced expression: The immunofluorescence data presented for WT and variant hSLC39A8 shows protein localization but does not sufficiently validate the expression induced by doxycycline at the gene or protein level. qPCR and Western blot are essential to confirm that both WT and variant hSLC39A8 are expressed comparably after doxycycline induction. These experiments would ensure that any functional differences observed are not due to varying expression levels.

We thank this reviewer for this point, we have now addressed this with immunoblot analysis in supplementary figure 2, in response to reviewer 1. In addition, we have included the QPCR analysis below. Other than the negative controls (non-induced and V1) all the variants showed amplification of the SLC39A8 mRNA, with PCR primers targeting the terminal region encompassing the HA and STREPII tag, normalized to GAPDH. V2 also did not amplify, however this was an early N-terminal mutation that may have affected expression at the mRNA level. (N=2)

- Metal ion transport assays: The authors claim to have measured metal transport for Cd²⁺, Zn²⁺, and Mn²⁺ using the FLIPR Calcium-5 Assay (for Cd²⁺), Zinpyr-1 dye (for Zn²⁺), and a cell viability assay (for Mn²⁺). However, none of these assays directly measure metal transport. The FLIPR Calcium-5 Assay measures calcium flux, not Cd²⁺ transport. Similarly, Zinpyr-1 measures Zn²⁺ levels in the cytosol but not Zn²⁺ transporter activity, and cell viability assays reflect general cell health, not Mn²⁺ transport. To accurately assess metal transport, the authors should employ specific uptake assays, such as radiolabeled or fluorescently tagged metal ions. Alternatively, the manuscript should be revised to clearly state that these methods assess changes in ion levels or viability, rather than direct transport activity.

We thank this reviewer for this point. The FLIPR calcium-5 assay is indeed a measure for calcium influx but it has also been used to indirectly measure Cd entry through a surrogate mechanism that was developed in this paper (<https://doi.org/10.1177/1087057114521663>). For all the other assays, it is true that they are indirect measures, and we have now made changes accordingly in the manuscript to indicate that what we see are changes in ion uptake or accumulation levels within the cell and not a direct implication on transport function. We also include this as a limitation in the discussion for cell-based assays (line 322-327).

- Conflicting reports on Ala391Thr variant expression: The manuscript cites conflicting reports regarding the expression of the Ala391Thr variant, but the current immunofluorescence data show no detectable protein expression of this variant in HEK 293 cells. This raises concerns about attributing the reduced Cd²⁺ and Zn²⁺ transport to functional defects when the lack of detectable expression could be the underlying cause. The authors should reconsider their

interpretation and clarify that the observed functional deficits may result from insufficient protein levels rather than inherent defects in the transporter. Further validation of expression levels is needed to resolve this issue.

We thank the reviewer for bringing this confusion to our attention. For V1, V2, V5, and V19, the introduction of the mutation affects the expression and therefore we don't see increased ion levels in these cells. The effect on the expression is therefore the mechanism by which these variants are loss of function variants. We have also validated our IF images with immunoblot analysis in supplementary figure 2 and now changed the text so that this is clearer (line 283).

- Figure 1A (Line 105): The variant number should be indicated in the topology diagram, and the location of the HA tag must be clearly shown.

We thank the reviewer for this suggestion. This has now been added to the diagram (Figure 1A).

- Supplementary Figure 1B (Line 113): The authors should include Supplementary Figure 1B in the main figures or at least show images of variants with different cellular localizations, such as those in the endoplasmic reticulum (ER). Showing only the WT localization is insufficient, as its plasma membrane localization is already established.

We thank the reviewer for this suggestion and have now provided zoomed in examples of each localization category in Figure 1C.

- Immunofluorescence for variants (Line 119): The authors state that variants V2, V5, and V19 affect ZIP8 expression, yet the immunofluorescence images (Supplementary Figure 1) show no detectable protein expression for these variants. The authors should address this discrepancy.

We thank the reviewer for bringing this confusion to our attention. This has been addressed in response to this reviewer's previous point, and we have made the text clearer (line 29, 112, 123, 241, 250).

Reviewer #3 (Comments to the Authors (Required)):

It is always a pleasure to review a manuscript that is exceptionally well-written and the data are unequivocal and on the cutting-edge of the field. Authors' abstract: The human SLC39A8 (hSLC39A8) gene encodes a plasma membrane protein SLC39A8 (ZIP8) that mediates the specific uptake of the metals Cd²⁺, Mn²⁺, Zn²⁺, Fe²⁺, Co²⁺, and Se⁴⁺. Pathogenic variants within hSLC39A8 are associated with congenital disorder of glycosylation type 2 (CDG type II) or Leigh-like syndrome. However, numerous mutations of uncertain significance are also linked to different conditions or benign traits. Our study characterized 21 hSLC39A8 variants and measured their impact on protein localization and transport for Cd²⁺, Zn²⁺, and Mn²⁺. We

identified three variants, including the highly pleiotropic Ala391Thr, with disrupted protein expression, five variants with high retention in the endoplasmic reticulum and 12 variants with localization to the plasma membrane. From the 12 variants with plasma membrane localization, we identified three with complete loss of transport function, and five with differential transport activity between metal ions. Further in silico analysis on protein stability identified variants that may affect the stability of homodimer interfaces. This study elucidates the variety of effects of hSLC39A8 variants on ZIP8 and on diseases involving disrupted metal ion homeostasis.

This Reviewer has offered various suggestions in the text of the pdf file - which are mostly trivial but which is grammatically more correct and/or less confusing to the General Reader. The biggest concern is trivial: with articles involving metals and/or organic compounds that participate in oxidation-reduction, it is clearer to the Reader to avoid "reduces" or "reduction" when synonyms such as "lowered", "decreases", "diminution" can be used instead.

Other than that, I wholeheartedly recommend publication of this article with the highest priority. The study is a masterpiece in its field.

We thank this reviewer for the generous and positive comments and have now incorporated all the suggestions and corrections throughout the manuscript.

January 6, 2025

RE: Life Science Alliance Manuscript #LSA-2024-03028R

Prof. Giulio Superti-Furga
CeMM Research Center for Molecular Medicine of the Austrian Academy of Sciences
Lazarettgasse 14
Vienna 1090
Austria

Dear Dr. Superti-Furga,

Thank you for submitting your revised manuscript entitled "Human genetic variants in SLC39A8 impact metal uptake specificity.". We would be happy to publish your paper in Life Science Alliance pending final revisions necessary to meet our formatting guidelines.

- please address Reviewer 2's remaining comments
- please be sure that the authorship listing and order is correct
- please add ORCID ID to the secondary corresponding author -- they should have received instructions on how to do so
- please add the Twitter handle of your host institute/organization as well as your own or/and one of the authors in our system
- please remove figures from the manuscript file and leave them uploaded separately.
- please add callouts for Figure S1A-D to your main manuscript text

A. FINAL FILES:

B. MANUSCRIPT ORGANIZATION AND FORMATTING:

Sincerely,

Reviewer #1 (Comments to the Authors (Required)):

The authors have taken into account all my comments satisfactory. The paper is now acceptable for publication.

Reviewer #2 (Comments to the Authors (Required)):

- The authors claim that some variants are not expressed at the protein level, implying a complete absence of detectable protein. However, they later report data on localization and metal uptake for these variants, which is contradictory. If a variant is truly not expressed at the protein level, there should be no data on its localization or functional activity, as these depend on the presence of the protein. This discrepancy requires clarification.
- Since the authors are comparing the metal uptake activities of different variants, normalization to protein expression is critical. Without proper normalization, variants with reduced protein expression could appear to have reduced functional activity when, in reality, the reduced uptake is simply due to lower protein abundance rather than intrinsic loss of function.
- The current title of the manuscript is misleading, as the authors do not directly measure metal uptake activity. Instead, they measure steady-state metal levels using ICP-MS and related assays. Therefore, the title should be revised to accurately reflect what was measured in the study.

Reply to Reviewer #2 (Comments to the Authors (Required)):

• **The authors claim that some variants are not expressed at the protein level, implying a complete absence of detectable protein. However, they later report data on localization and metal uptake for these variants, which is contradictory. If a variant is truly not expressed at the protein level, there should be no data on its localization or functional activity, as these depend on the presence of the protein. This discrepancy requires clarification.**

Reply: All the cells with variants that result in no expression (V2, V5, V19) are already reported as no expression, and also no localization to PM or ER. However, these variants were used in other assays where they offered insight as controls, because if there is no expression there should be no cellular uptake or increase in steady-state metal levels. This is reported as such in the manuscript. We hope that this reply makes the use of those variants clear.

• **Since the authors are comparing the metal uptake activities of different variants, normalization to protein expression is critical. Without proper normalization, variants with reduced protein expression could appear to have reduced functional activity when, in reality, the reduced uptake is simply due to lower protein abundance rather than intrinsic loss of function.**

Reply: Also protein expression does not accurately depict these uptake activities, as expression does not differentiate between PM or ER proteins, nor accurately dimer or monomer levels. To address this, and after the first round of revisions, we changed the wording to indicate that the variants affect cellular uptake or metal levels and not directly SLC transport activity. Therefore, the consequence of the variant may be due to a number of reasons including amount of functional SLC39A8. But pinpointing exactly how much level of functional SLC is beyond our abilities. Whether the variant result in reduced expression, the consequence is still reduced metal uptake. Ultimately, we are reporting the consequence of the mutation.

• **The current title of the manuscript is misleading, as the authors do not directly measure metal uptake activity. Instead, they measure steady-state metal levels using ICP-MS and related assays. Therefore, the title should be revised to accurately reflect what was measured in the study.**

Reply: We thank the reviewer for this suggestion. We have now changed the title to reflect this.

January 15, 2025

RE: Life Science Alliance Manuscript #LSA-2024-03028RR

Prof. Giulio Superti-Furga
CeMM Research Center for Molecular Medicine of the Austrian Academy of Sciences
Lazarettgasse 14
Vienna 1090
Austria

Dear Dr. Superti-Furga,

Thank you for submitting your Research Article entitled "Human genetic variants in SLC39A8 impact uptake and steady-state metal levels within the cell.". It is a pleasure to let you know that your manuscript is now accepted for publication in Life Science Alliance. Congratulations on this interesting work.

DISTRIBUTION OF MATERIALS:

Again, congratulations on a very nice paper. I hope you found the review process to be constructive and are pleased with how the manuscript was handled editorially. We look forward to future exciting submissions from your lab.

Sincerely,
